# Boosting Clear Cell Renal Carcinoma-Specific Drug Discovery Using a Deep Learning Algorithm and Single-Cell Analysis

**DOI:** 10.3390/ijms25074134

**Published:** 2024-04-08

**Authors:** Yishu Wang, Xiaomin Chen, Ningjun Tang, Mengyao Guo, Dongmei Ai

**Affiliations:** School of Mathematics and Physics, University of Science and Technology Beijing, Beijing 100083, China; wangyishu@ustb.edu.cn (Y.W.); m202110683@xs.ustb.edu.cn (X.C.); 42022086@xs.ustb.edu.cn (N.T.); m202210694@xs.ustb.edu.cn (M.G.)

**Keywords:** single-cell RNA sequencing, ccRCC, tumor microenvironment heterogeneity, deep learning algorithm, specific drug discovery, EPAS1/HIF-2α

## Abstract

Clear cell renal carcinoma (ccRCC), the most common subtype of renal cell carcinoma, has the high heterogeneity of a highly complex tumor microenvironment. Existing clinical intervention strategies, such as target therapy and immunotherapy, have failed to achieve good therapeutic effects. In this article, single-cell transcriptome sequencing (scRNA-seq) data from six patients downloaded from the GEO database were adopted to describe the tumor microenvironment (TME) of ccRCC, including its T cells, tumor-associated macrophages (TAMs), endothelial cells (ECs), and cancer-associated fibroblasts (CAFs). Based on the differential typing of the TME, we identified tumor cell-specific regulatory programs that are mediated by three key transcription factors (TFs), whilst the TF EPAS1/HIF-2α was identified via drug virtual screening through our analysis of ccRCC’s protein structure. Then, a combined deep graph neural network and machine learning algorithm were used to select anti-ccRCC compounds from bioactive compound libraries, including the FDA-approved drug library, natural product library, and human endogenous metabolite compound library. Finally, five compounds were obtained, including two FDA-approved drugs (flufenamic acid and fludarabine), one endogenous metabolite, one immunology/inflammation-related compound, and one inhibitor of DNA methyltransferase (N4-methylcytidine, a cytosine nucleoside analogue that, like zebularine, has the mechanism of inhibiting DNA methyltransferase). Based on the tumor microenvironment characteristics of ccRCC, five ccRCC-specific compounds were identified, which would give direction of the clinical treatment for ccRCC patients.

## 1. Introduction

At present, renal cell carcinoma (referred to as renal carcinoma) is a high degree of malignancy in the urinary system and is also one of the most common tumors; it originates in the renal parenchyma urotubular epithelial system of malignant tumors, also known as renal adenocarcinoma, accounting for 80% to 90% of renal malignancies. It is the most common and deadly tumor of the urinary system and accounts for 2% to 3% of adult malignant tumors. Its morbidity and mortality are expected to increase, with more than 300,000 deaths estimated by 2040 [1,2]. Clear cell renal carcinoma (ccRCC) is the most common type of renal carcinoma [3,4], with many immunologically distinct tumor types [5,6]. The treatment for ccRCC includes partial or radical nephrectomy and ablation (radiation, freezing, or microwave ablation) [7,8], as early immunotherapy with immunostimulants and cytotoxic chemotherapy drugs has limited efficacy, with approximately a 10% response rate [9].

With the development of single-cell sequencing technology, a number of scRNA-seq data analyses have been used to comprehensively characterize the cellular composition and transcriptional states of ccRCC [5,10]. Many common mutations of ccRCC have been documented, such as the inactivation of the VHL gene; specific tumor immune microenvironment characteristics, such as proliferative CD4+ T cells; and ccRCC-specific CD45+ lymphoid and myeloid cells [11,12,13,14,15]. Meanwhile, emerging targeted therapies, such as the HIF-2 small molecule inhibitors PT2385 and PT2399 [16], which are inhibitors of the HIF2-VEGF [17] and NRF2 signaling pathways [18], have revolutionized the treatment of ccRCC in recent years; more than 10 kinds of treatment, such as hemangiogenesis inhibitors, rapamycin target egg (mTOR) inhibitors, and immune checkpoint inhibitors [19,20,21], have been used. Nevertheless, targeted therapy is prone to drug resistance, and immunotherapy is limited by tumor heterogeneity. Therefore, exploring new drug targets to reverse drug resistance and improve patients’ survival and quality of life is highly valuable.

Notably, the occurrence, growth, and metastasis of cancers are highly correlated to the structure and function of their tumor cells and tumor microenvironment (TME), which is a complex cellular network shaped by heterogeneous malignant cells and interacting immune and non-immune cells [22]. Therefore, it is crucial to systematically determine the specific characteristics of the ccRCC TME to effectively explore new candidate drugs for targeted therapy and immunotherapy. However, conventional strategies for determining the properties of a given molecule usually require a series of complicated biochemical reactions and chemical structural calculations. With the increase in the number of known chemical compounds and complex protein structures, drug selection using experimental methods and traditional machine learning algorithms has become impossible. Drug development is a time-consuming and costly process.

In this study, we used the scRNA-seq data of six patients in the GEO database (https://www.ncbi.nlm.nih.gov/geo/, accessed on 20 December 2023) to describe the landscape of the TME in ccRCC. According to the distinct transcriptomic patterns associated with TME components (such as T cells, TAMs, and ECs), 150 candidate genes related to the development and pseudo-differentiation of ccRCC were identified and screened for tumor-related regulatory factors [23]. Subsequently, through comparison with the TF database, three key TFs associated with tumor functional states and the immune microenvironment were identified as candidate ccRCC inhibitor targets. After an analysis of their protein structures, EPAS1/HIF-2α was selected for virtual screening as the drug target of an anti-ccRCC therapy by one neural network-based molecular feature model (DMPNN) and an optimal machine learning classifier (GBDT), DMPNN + GBDT [24] to predict potential compounds specific to EPAS1 protein. Then, the Schrodinger Maestro 11.4 software was used for virtual screening, and PyMol was used for 3D mapping, which resulted in five potential ccRCC-specific compounds, including two FDA-approved drugs (flufenamic acid and fludarabine), one endogenous metabolite, one immunology/inflammation-related compound, and one inhibitor of DNA methyltransferase (N4-methylcytidine, a cytosine nucleoside analogue that, like zebularine, has the mechanism of inhibiting DNA methyltransferase, as well as potential anti-metabolic and anti-tumor activities). Considering that the FDA-approved drugs have experienced strict drug metabolic reactions and other chemical and clinical experiments, they would be more suitable for the direction of rapid new use of old drugs. Finally, we performed the molecular dynamics to further prove the mechanism of interactions between protein EPAS1 and these two drugs.

The phenotypic and functional profiles of different immune cells in the tumor microenvironment are well known to influence prognosis and tumor process. Simultaneously, ccRCC tumors have many counterintuitive immune findings that impact therapy of ccRCC patients. In this study, our scRNA-seq data analysis revealed a ccRCC tumor-specific regulatory signature, and three key TFs associated with TME states were identified. Moreover, with the implementation of a deep learning algorithm and virtual screening methods, five potential compounds were discovered, including two FDA-approved drugs. Therefore, our work provides one important insight into exploring ccRCC tumor progress-related hub TFs and the corresponding specific potential drugs, which would provide a novel therapeutic approach with potential for treating ccRCC tumors.

## 2. Results

### 2.1. The Landscape of Cell Type Clustering in ccRCC According to an scRNA-seq Analysis

The transcriptomic data downloaded from the GEO database underwent base-calling, alignment, filtering, and normalization. We performed further quality control (QC) by retaining high-quality cells in each of the following metrics using the Seurat (v4.0.3) [25] R package: cells with more than 200 genes, cells with more than 400 UMI counts, and cells with a percent mitochondrial read count less than 20%. The general description of the methods and materials used in this study is shown in the following workflow (Figure 1).

After QC was conducted using Seurat, 31,625 high-quality single-cell transcriptome data points were obtained from the ccRCC patients. The detailed description of methods and algorithms is exhibited in the Methods and Materials section. Based on their marker genes (FGS5, NDUFA4L2, IGFBP3, PLVAP, APOE, C1QB, CCL5, KLRB1, LST1, LYZ, COL1A1, TIMP1, IGLC2, and IGLC3) (expression levels of these marker genes are shown as one heatmap in Appendix A), these cells were categorized into five major subtypes, namely ccRCC tumor cells, TME-related cells, T cells, ECs, CAFs, and TAMs (Figure 2A). An unbiased clustering analysis based on scRNA-seq revealed different cellular subtypes of the ccRCC lesions. The marker genes for different subclusters are presented in a heatmap (Figure 2B). Moreover, the distribution of different types of cells in these six samples is shown in Appendix A.

When investigating the heterogeneous tumor microenvironment (TME) in ccRCC, malignant cells were shown to interact with both immune and non-immune cells to shape this complex ecosystem. Figure 2 reveals a large number of tumor-associated macrophages, endothelial cells, T cells, and fibroblasts in the ccRCC TME. We subsequently isolated these strains for further investigation.

### 2.2. The Four Kinds of Subsets Identified in Macrophage Cells

As an important cellular component of innate immunity, macrophages can respond to invading pathogens, phagocytose, and digest pathogens and also participate in the removal of damaged and senescent cells, simulating adaptive immunity through antigen presentation [26]. However, in addition to fighting disease, macrophages are also associated with the occurrence and development of chronic diseases, autoimmune diseases, and tumors. Macrophages that infiltrate tumor tissue or aggregate in solid tumor microenvironments are known as tumor-associated macrophages (TAMs) [27]. In this study, two kinds of TAM subtypes, one classical monocyte cell type and one type of unpolarized macrophage, and a small number of conventional type 1 dendritic cells (cDC1s) were found (Figure 3A). Moreover, the evolutionary trajectory of these clusters was reconstructed from their cell differentiation order (Figure 3B) via Monocle2; most classical monocytes, most macrophages, and the cDC1s were at the early stage of development, while the M1 and M2 TAM subtypes were present and at two differentiation stages. These findings demonstrated that the TAMs in ccRCC tumor cells are differentiated from monocytes. The two phenotypes of TAMs have different “characteristics”; M1 macrophages have high antitumor, antipathogen, phagocytic, and proinflammatory effects, while M2 macrophages play an anti-inflammatory role, participating in tissue remodeling, healing, and promoting tumor growth, and are associated with tumor progression and immunosuppression. The three genes that are most critical to the development of these macrophages’ trajectories are ACTB, CAPG, and STXBP2 (Figure 3C). Subsequently, the top 50 genes involved in developing these subtypes, which varied according to their pseudotime development cluster, were further clustered according to their initial expression patterns and terminal patterns (Figure 3D).

### 2.3. Six Subtypes of Endothelial Cells in ccRCC

The stromal populations of TME endothelial cells (ECs) have been demonstrated to modulate inflammation by regulating the trafficking, activation status, and function of immune cells [28]. Kidney Ecs represent a particularly heterogeneous population with potentially immunomodulatory effects [29]. ccRCCs are highly vascularized tumors with disorganized vasculature, endothelial cells, and pericytes [30]. Recent single-cell studies of ccRCCs have revealed specific EC subtypes that express gene signatures indicative of phagocytosis or scavenging, antigen presentation, and immune cell recruitment, with several subclusters or more [30,31,32,33]. In this study, we also identified five EC subtypes and one pro-EC subtype (Figure 4A). The reconstructed evolutionary trajectories of these EC subclusters are shown in Figure 4B, along with their developmental order; pro-EC and EC3 cells were almost at their earliest stage, while after three passages, they differentiated into EC1 and EC5 cells throughout the end stage of the pseudotime order. EC3 cells highly expressed CCL5, and pro-EC cells significantly expressed SPP1, whilst EC1 and EC5 highly expressed RND1 and ENPP2 (Figure 4C).

CCL5 is a chemokine that forms a superfamily of secreted proteins involved in immunomodulatory and inflammatory processes and acts on blood monocytes, memory T helper cells, and eosinophils [34]. SPP1 (secreted phosphoprotein 1) is a protein-coding gene. The diseases associated with SPP1 include pediatric systemic lupus erythematosus and dentin dysplasia, and its related pathways are the integrin and ERK signaling pathways. More notably, SPP1 has been demonstrated to be involved in immune regulation, cell survival, and tumor progression [35]. Therefore, pro-EC and EC3 play important roles in the early stages of the development of EC differentiation.

At the end stage of EC development, the genes RND1 and ENPP2 were highly expressed (Figure 4B). RND1 is a novel membrane-associated protein that inhibits RhoA activation through Plxnb1 and promotes the expression of the pro-inflammatory cytokines IL-6 and TNF-α, thereby counteracting intracellular calcium fluctuations [36]. In addition, ENPP2 is an essential protein for normal development, and changes in its expression are associated with embryonic and neural development, migration, invasion, differentiation, proliferation, angiogenesis, and survival [37]. Therefore, EC1 and EC5 may play a role in vascular development and remodeling, which is consistent with our finding that EC5 is the end stage of ECs, where the vessels in lesions are euangiotic. The top 50 genes that were significantly expressed in the differentiation of EC subclusters’ pseudodevelopment were chosen as candidate genes.

### 2.4. The Diversity of Fibroblasts (Including Multiple CAFs) in ccRCC, Revealed by scRNA-seq

Cancer-associated fibroblasts (CAFs), which are widely present in tumor cells and more than 50% of stromal cells, are a group of activated fibroblasts that can secrete a variety of active factors to regulate the occurrence, development, and metastasis of tumors [38]. Four subtypes of CAFs were identified in ccRCCs: three different CAF subclusters and one fibroblast cluster (Figure 5A). Moreover, a trajectory analysis revealed that the fibroblasts were at an early stage of development, while the CAF-3 and CAF-2 cells were differentiated into two groups (Figure 5B) with different transcriptional characteristics (Appendix A).

The CAF3 cluster highly expressed the genes Lumican (LUM) and COL1A1 (Appendix A); LUM may regulate collagen fibril organization, circumferential growth, corneal transparency, epithelial cell migration, and tissue repair [35], while COL1A1 provides instructions for making part of a large molecule called type I collagen and is involved in the epithelial–mesenchymal transition. Furthermore, COL1A1 is highly expressed in various cancers and regulates various cellular processes, including cell proliferation, metastasis, and apoptosis [39] (Figure 5C). Therefore, CAF3 is highly correlated with tumor progression.

On the contrary, CAF-2 cells highly expressed CCL5 and NKG7. CCL5 is a chemokine gene that functions as a chemoattractant for blood monocytes, memory helper T cells (CD4+ T cell), and eosinophils [40,41]. New studies have suggested that the NK cell granule protein NKG7 is essential for NK and CD8+ T cells’ cytotoxic degranulation and CD4+ T cells’ activation and proinflammatory responses [40,41]. Furthermore, in malignant tumors, CAFs promote tolerance by recruiting and inhibiting regulatory T cells through the secretion of CCL2, CCL5, etc. [42] (Figure 5C). Therefore, the CAF-2 subtype is associated with a chronic inflammatory environment and immune responses. Subsequently, this subgroup of the top 50 most highly expressed genes in different pseudotime development clusters were also regarded as candidate genes.

### 2.5. Diversity of T Cell Characteristics in the ccRCC Immune Microenvironment

T cells are immune cells with tumor-killing characteristics that play vital roles in parts of active immunity, including cell-mediated and, to some extent, humoral immunity [43]. The most common and well-known T cells are CD4+ T cells and CD8+ T cells. In this study, the T cells in the ccRCC cell population were identified as CD8+ T cells 1, CD8+ T cells 2, CD8+ T cells 3, and CD4+ T cells (Figure 6A, Appendix A). However, the expression of T cells in these six ccRCC samples was far below that of other cells, except for the sample ccRCC1 (Figure 6B), which was from a stage 3 tumor, while the other five samples were all stage 2. A survival analysis of patients stratified according to their expression of the marker genes of T cells in the TCGA dataset [44] suggested that the prognosis of patients with high T cell infiltration was poor (Figure 6C).

According to their gene expression characteristics, three types of T cells were identified: resident memory CD8+ T cells (CD8 Trm), naive T cells (T naive), and regulatory T cells (T regs) (Appendix A). As a major immunosuppressive subset of CD4+ T cells, Treg cells have been found to substantially infiltrate many solid tumors [45,46]. A high frequency of Treg cells is primarily associated with worse clinical outcomes in many kinds of tumors, such as melanoma, breast, lung, cervical, gastric, renal, endometrial, and ovarian cancers [47,48,49]. Therefore, we considered the main characteristic genes of Treg cells and CD4 + T cells—IL2, CD25, IL10, TGF-β, IL35, and FOXp3—taking these into account as candidate genes during the next step of selecting TFs.

### 2.6. TFs and Their Related Significantly Active Compounds

To understand the underlying function of the heterogeneous TME, we investigated the TF regulation programs that regulate the development of the TME in ccRCC. Using the transcription factor (TF) database described by Lambert et al. [50], we filtered the TFs on our candidate gene list, which was compiled from our above analysis of the TME’s heterogeneous cell types. Three TFs (EPAS1, HES1, and ID3) were filtered from the candidate gene list, which show a substantial increase in both their gene expression and activity in the development of their corresponding pseudotime clusters. Although these three TFs have three-dimensionally analyzed protein structures, the protein fragments of HES1 and ID3 were too short and disordered for virtual screening. Therefore, we focused on the endothelial PAS domain protein 1 (EPAS1), which had an appropriate-length fragment: 239–350. The workflow of the candidate drugs targeting EPAS1, selected using virtual screening, is shown in Appendix A.

### 2.7. EPAS1/HIF-2α Is Highly Associated with ccRCC and May Be a Clinical Biomarker and Drug Target

Endothelial PAS domain protein 1 (EPAS1), also named hypoxia-inducible transcription factor 2 (HIF-2α), is a basic helix-loop-helix/PAS domain transcription factor, expressed most abundantly in highly vascularized organs [51]. HIFs, as a major regulator of homeostasis, regulate the response to hypoxia through the transcriptional activation of multiple genes, including the erythropoietin (EPO) gene and the vascular endothelial growth factor (VEGF) gene [52]. Notably, the mutational inactivation of VHL is the earliest genetic event in the majority of clear cell renal cell carcinomas (ccRCCs), while HIFs could be ubiquitinylated by this tumor suppressor protein (VHL), leading to the accumulation of HIF-1α and HIF-2α and regulating the development and inflammation of ccRCC [51]. Furthermore, many studies have found that HIF-2α is induced by T helper 2 (Th2) cytokines during M2 macrophage polarization, and a lack of HIF-2α in the myeloid lineage will result in decreased TAM infiltration and alleviated tumor progression [53]. A loss of HIF-2α also increases neutrophil apoptosis and reduces neutrophilic inflammation [54]. In conclusion, EPAS1/HIF-2α has close relationship with the occurrence and development of ccRCC, as the prognostic marker with the highest expression in renal carcinoma [55], and an important role in the ccRCC-related pathway, as shown in WikiPathways (https://www.wikipathways.org/pathways/, accessed on 5 January 2023) (Figure 7A).

According to information from Uniprot (https://www.uniprot.org/, accessed on 20 December 2023), the protein EPAS1/HIF-2α consists of one basic helical ring helix domain (PAS), two PER-ARNT-SIM domains (A and B), one oxygen-dependent degradation domain (ODDD), and two transcriptional activation domains (N-TAD and C-TAD), where two proline residues (P405 in ODDD and P531 in N-TAD) and asparagine residues (N847 in C-TAD) are hydroxylated during physiological oxygen tension, regulating the stability and activity of the HIF-2α protein [56,57], as shown in Figure 7B, which is in reference to [56]. In particular, HIF-2α’s PAS-B domain contains a large internal cavity that can be used to identify small molecular ligands.

### 2.8. Five Compounds Were Identified to Target the TF EPAS1

After the molecular feature extraction of the deep learning algorithm DMPNN + XGBoost and its molecular docking prediction and docking scores of the target protein “EPAS1”, the five compounds with the top-ranked docking scores were selected. Compound 1, HY-B1673, an endogenous metabolite, can form four hydrogen bonds with the EPAS1 protein, where three hydroxyl groups in its pyranoid ring form a hydrogen bond with HIS248/SER246/TYR307, respectively (Figure 8A). Compound 2, phenyl β -D-glucopyranoside, with anticancer and anti-inflammatory activities, can inhibit the production of nitric oxide, the expression of COX-2, and the nuclear translocation of NF-kB [58]. It can form two hydrogen bonds with EPAS1, with TYR-281 and CYS-339, respectively (Figure 8B). Compound 3 and compound 4 are two FDA-approved drugs, flufenamic acid and fludarabine, respectively. Flufenamic acid is a non-steroidal anti-inflammatory agent which can inhibit the activity of COX, regulate ion channels, and block the chloride and L-type Ca2+ channels. Furthermore, flufenamic acid can inhibit TEAD function and TEad-YAP-dependent processes such as cell migration and proliferation [59,60]. Moreover, fludarabine (F-ara-A; NSC 118218) is a DNA synthesis inhibitor and a fluorinated purine analogue with antitumor activity towards lymphoproliferative malignancies, which can inhibit the activation of STAT1 and STAT1-dependent gene transcription induced by cytokines. There have been more than thirty studies of fludarabine over the past few years [61,62]. The docking models in Figure 8C,D demonstrate that these two drugs can target the TF EPAS1. Besides their hydrogen bonding ability, they can also generate π–π interactions. The last compound is a cytosine nucleoside analogue, which has mechanisms to inhibit DNA methyltransferase (e.g., zebularine) and potential antimetabolic and antitumor activities [63]. It also has a comparatively stable binding pocket with the protein EPAS1 through more than four potential hydrogen bonds (Figure 8E). Studies have demonstrated that EPAS1 is regulated by DNA methyltransferases (DNMTs) in non-small cell lung cancer (NSCLC) [64].

Because molecular docking is rigid docking and the flexibility of protein structure cannot be considered at present, in order to further prove the degree and stability of binding between compounds and proteins, we performed the 25 ns molecular dynamics of EPAS1 P and the two FDA drugs: flufenamic acid and fludarabine. Figure 9A,E exhibit the root mean square deviation curve (RSMD) results, which is an index to determine the stability of protein ligand complexes, The fluctuation range of the RMSD curve is within 3 Å all throughout. Furthermore, the fluctuation of amino acid residues on the main chain of complex protein is basically within 1.5 Å, and there is no large fluctuation; only the end amino acid residues in the flexible region of the end produced reasonable fluctuations greater than 3Å (Figure 9B,F). On the other hand, the conformation of small molecules in the complex fluctuated were all within 1.5 Å (Figure 9C,G). In the end, in order to study the interaction between small molecules and each residue in the protein pocket, the interactions involved in the simulation locus were counted residues and the interactions in which each residue participates, including hydrogen bonding, salt bridges, water bridges, and hydrophobic interactions. In the complex of flufenamic acid, the protein residues HIS-248,SER-249, Ph-254,TYR-281, and HIS-293 had hydrogen bonding, salt bridge, water bridge, or hydrophobic interaction with small molecules in about 70% of the simulated tracks, indicating that these residues had a high contribution to binding free energy (Figure 9D). In the complex of fludarabine, protein residues ASN_341,TYR_307 and CYS_339 have stable hydrogen bond interactions with small molecules HY-B0069 and contribute significantly to the free energy of binding. In addition, SER_304 and TYR_281 have hydrogen bond interactions with small molecules in about 40% of the trajectory, which is also important (Figure 9H).

## 3. Materials and Methods

### 3.1. Dataset Preparation

A total of 31,625 single-cell transcriptome images acquired from six ccRCC samples were downloaded from the GEO database (https://www.ncbi.nlm.nih.gov/geo/, accessed on 20 January 2024) with the following GEO identifiers: GSM4735364, GSM4735366, GSM4735368, GSM4735370, GSM4735372, and GSM4735374. The ChEMBL database [50] (version 32) and the active compound library of the MCE (https://www.medchemexpress.cn/, accessed on 20 January 2024) database, which contains 50,000 small molecules with well-defined reported activities and well-defined targets, including bioactive compound libraries, anticancer natural product libraries, FDA-approved and pharmaco-pharmaceutical drug libraries, a natural product library, and human endogenous metabolite compound library, were used to select clinical targets related to “EPAS1” and “HIF-2α” and train the integrated deep learning and machine learning algorithm.

### 3.2. Molecular Feature Extraction Model and Virtual Screening Software

We downloaded the simplified molecular input line entry system56 (SMILES) [65] of compounds from the ChEMBL database and adopted the RDKit tool [66] to process these SMILES-encoded compounds to obtain their molecular graphs and Morgan fingerprints [67]. The DMPNN + GBDT model [24] was used to extract their molecular features and predict active targets. Subsequently, Schrödinger Maestro 12.8 (https://www.schrodinger.com, accessed on 20 January 2024) was used for the precise docking of their ligands and receptors.

Protein preparation: download the 3D structure of human EPAS1 (PDB ID: 3F1O) from the PCSB PDB website (https://www.rcsb.org/, accessed on 20 January 2024). The Protein Preparation Wizard module hydrogenates the protein and removes the water molecule and B chain. We then performed energy optimization (OPLS2005 force field, RMSD = 0.3 Å). The treated protein is used in Receptor Grid Generation mode Block to generate a grid file centered on the pocket where ligand THS-044 is located and set the box size to 20 Å × 20 Å × 20 Å.

Compounds preparation: the 2D format of HY-L001P MCE Bioactive Compound Library Plus (including 19.1 K compounds) was processed by Schrodinger software LigPrep Module for hydrogenation and energy optimization. Output 3D structure was used for virtual filtering.

Molecular docking: the Virtual Screening Workflow module was used to perform virtual screening and import the prepared compounds. The Glide module was used for molecular docking, that is, the receptor and ligand molecules are paired with each other through geometric and energy matching. First, the small molecule compounds prepared in the database were screened using the high-throughput screening (HTVS) mode in the Glide module, and the top 15% of small molecule compounds were selected for the second round of screening using the standard (SP) mode. The first 15% of the scoring value is then selected in high-precision (XP) mode for the third round of screening to obtain the ranking of small molecule compounds. The higher the absolute value of the docking_score, the stronger the binding force between the compound and the protein.

### 3.3. Cell Clustering Analysis, Visualization, and Annotation

A Seurat object with gene expression data was imported into the Seurat (v2.3.0) R toolkit with the Read 10× function [25]. The normalizeData function in the Seurat package of the R toolkit, version 5.0.1, was used to normalize its raw counts with the command LogNormalize. The cells were visualized through two-dimensional uniform manifold approximation and projection (UMAP) [68]. Cell type annotation was performed using reference-based annotation and the R package “Single R”. After dimension reduction for each dataset, single cells were annotated with known cell types using Single R and reference ccRCC marker genes [69].

### 3.4. Functional Enrichment Analysis

GO and KEGG enrichment analyses were performed using the R package ClusterProfiler4.0 [70] for the functional enrichment analysis of the differently expressed genes (DEGs) in each cluster. An adjusted of Padj< 0.05 was used as the screening criterion for significant enrichment in all enrichment analyses.

## 4. Discussion

Clear cell renal carcinoma (ccRCC) is the most common type of renal carcinoma and is one of the most immunologically distinct tumor types due to its high response rate to immunotherapies. With the development of single-cell sequencing technology, a number of scRNA-seq data analyses have been used to comprehensively characterize the cellular composition and transcriptional states of ccRCC. Although many mutations and abnormal pathways have been identified as targeted therapy and immunotherapy directions, drug resistance and individual therapeutic differences due to tumor heterogeneity lead to a high proportion of failures in treatment of ccRCC. In this study, we tried to discover the tumor-related transcription factors that regulate the occurrence and development of ccRCC by exploring the characteristics of its tumor microenvironment. Then, an integrated deep learning model that combined a graph neural network model and a machine learning algorithm was used to extract molecular features to select active compounds that targeted candidate TF proteins. Finally, our virtual screening found the five most likely compounds to dock with the TF EPAS1. Notably, EPAS1/HIF-2α has been used as a therapy target in the past [71,72], but several inhibitors have revolutionized the treatment of ccRCC. EPAS1/HIF-2α has been proven to have relationship with the TME [73]. Moreover, EPAS1/HIF-2α has also been demonstrated to regulate or take part in many tumor-related pathways, such as the VEGF and PI3K-Akt pathways [74,75]. Furthermore, the degradation of HIF-2α is induced by products of the VHL gene [76]. Consequently, the deficiency of these products can cause many tumors. 

Therefore, in this sense our methodology of identifying key TFs in regulating TME of malignant tumors and the implementation of deep learning and virtual screening for filtering candidate compounds both have practical significance. Moreover, in 2019, William G. Kaelin Jr, Sir Peter J. Ratcliffe, and Gregg L. Semenza were awarded the Nobel Prize because of their contribution in the research of protein VHL-1, which is a double-edged sword; while being essential for embryonic development, oxygen balance, and other processes, it is also a culprit for promoting cancer occurrence, proliferation, and metastasis. It is precisely because the HIF pathway has a key role in the development of tumor resistance to different treatment modalities that a higher expression of the HIF molecule is associated with poor prognosis. EPAS1 is an important hub protein in tumor development and still has research value for anti-tumor targeted therapies. Recently, one new highly specific and well-tolerated HIF-2α inhibitor, belzutifan, became FDA-approved for the treatment of nonmetastatic renal cell carcinomas and pancreatic neuroendocrine tumors [77].

However, the drug research and development would cost huge amounts of time and financial resources. One investigation in 2016 based on a random sample of 68 drugs (both chemical and biological) developed internally by 10 multinational companies calculated the pre-tax cost of each new drug (taking into account the cost of drug failures) at 2.826 billion dollars [78]. Therefore, computer-assisted drug design in preliminary screening has become more and more important and necessary for new drug discovery. On the other hand, in pharmaceutical research, using conventional drugs in new ways is one important strategy to simplify the new drug discovery process, by application of high-throughput virtual screening technology, target docking screening technology, etc., to discover new mechanisms from drugs already on the market [79], which would greatly shorten the development time and research costs.

In this study, in order to screen specific potential drugs that target EPAS1 more efficiently, an integrated deep learning model and virtual screening docking software were used on a high-throughput database (more than 50,000 compounds), resulting in five potential compounds, including two FDA-approved anti-cancer drugs (flufenamic acid and fludarabine). The docking models have demonstrated the high likelihood of these compounds binding to EPAS1. The new mechanism prediction of flufenamic acid and fludarabine would exempt the cost of existing toxicological and pharmacokinetic evaluations in drug research. Therefore, in the end of our study, the molecular dynamics were adopted for these two drugs, which predicted the behavior and structural properties of molecular systems by studying the motion and interaction of molecular systems. These results further provided theoretical basis for clinical researchers and drug developers in using flufenamic acid and fludarabine for the therapy of ccRCC tumors.

However, there have been several studies applying scRNA-seq analysis to generate transcriptional landscapes of ccRCC and identify tumor-specific regulating key TFs [80,81]. Our study identified the tumor-specific regulatory programs, i.e., the key TFs through the whole landscape of TME in ccRCC tumors. Furthermore, the deep learning algorithms and virtual screening boosted new drug discovery for the therapy of ccRCC and also highlighted one possible direction for medicinal clinical treatment with no necessary pharmacokinetic experiments. Therefore, this study provided a theoretical basis for targeted ccRCC therapy for clinical researchers and the direction of new use of old drugs for ccRCC tumors, therefore greatly improving drug research efficiency.

Our study has limitations, including the small total number of samples (*n* = 6), the rarity of T cells, the need for further experiments, such as surface plasmon resonance (SPR) assays and microscale thermophoresis (MST) assays, and the limitation of the dimensionality reduction UMAP, which is a standard practice for filtering noise and identifying relevant features in large-scale data analyses. However, many studies have demonstrated that extreme dimension reduction inevitably induces significant distortion [82]. Moreover, T. Chari et al. examined the practical implications of low-dimensional embedding of single-cell data [82]. However, taken together, we provided a transcriptional TME map of ccRCC tumor and discovered the key TFs and corresponding potential specific drugs in the hope of gaining insight into biomarkers and therapeutic targets in ccRCC, which would greatly save the cost of new drug research and development.

## 5. Conclusions

In this study, we constructed a comprehensive landscape of the ccRCC TME, including the heterogeneous subclusters of its tumor-associated macrophages, endothelial cells, cancer-associated fibroblasts, and T cells. Based on the differential typing of these key TME characteristics and the computation of an integrated deep learning model, the pivotal TF protein EPAS1/HIF-2α was identified, which regulates the expression of the genes involved in adjusting the mechanisms related to hypoxia, such as angiogenesis or apoptosis and tumor growth and invasion. Finally, we screened five candidate compounds, including two FDA-approved anti-cancer drugs and one inhibitor of DNA methyltransferase, which all form an important theoretical foundation for the new use of old drugs and the development of new pharmaceuticals for the treatment of ccRCC to save drug research costs and avoid drug resistance.

Instantly, the cancer treatment was heavily limited by drug resistance, tumor heterogeneity, and so on. In this study, we found one important target, EPAS1/HIF-2α, which is an important hub protein in tumor development and still has research value for anti-tumor targeted therapies. The HIF pathway has a key role in the development of tumor resistance to different treatment modalities. Therefore, EPAS1 is much more likely to be one potential therapy target to solve the old drug resistance in ccRCC clinical treatment. Moreover, through high-throughput virtual screening, two FDA-approved drugs were filtered as candidate anti-ccRCC drugs, which highlights one possible direction for new uses of conventional drugs.

## Figures and Tables

**Figure 1 ijms-25-04134-f001:**
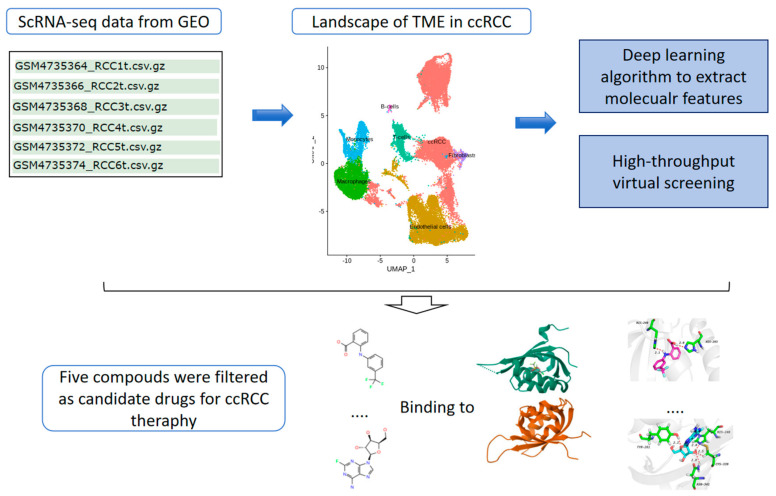
The workflow of methods and process used in this study.

**Figure 2 ijms-25-04134-f002:**
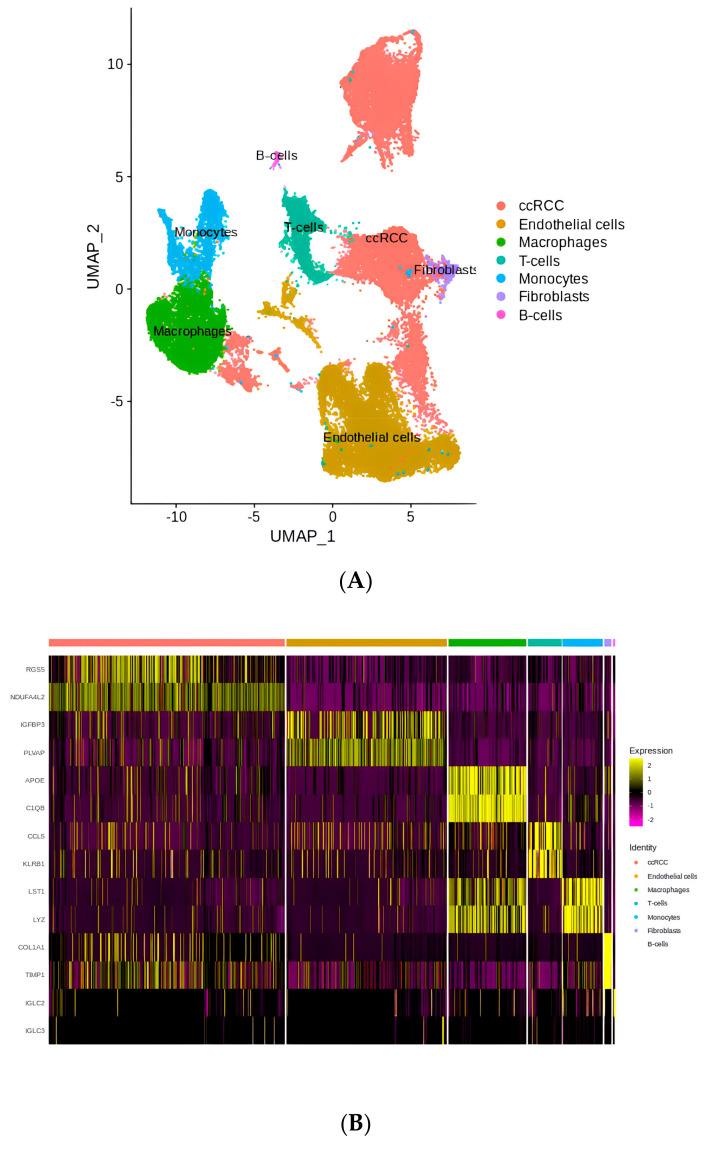
(**A**) The landscape of cell clusters determined via scRNA−seq. (**B**) Heatmap of the marker genes in different cell subclusters.

**Figure 3 ijms-25-04134-f003:**
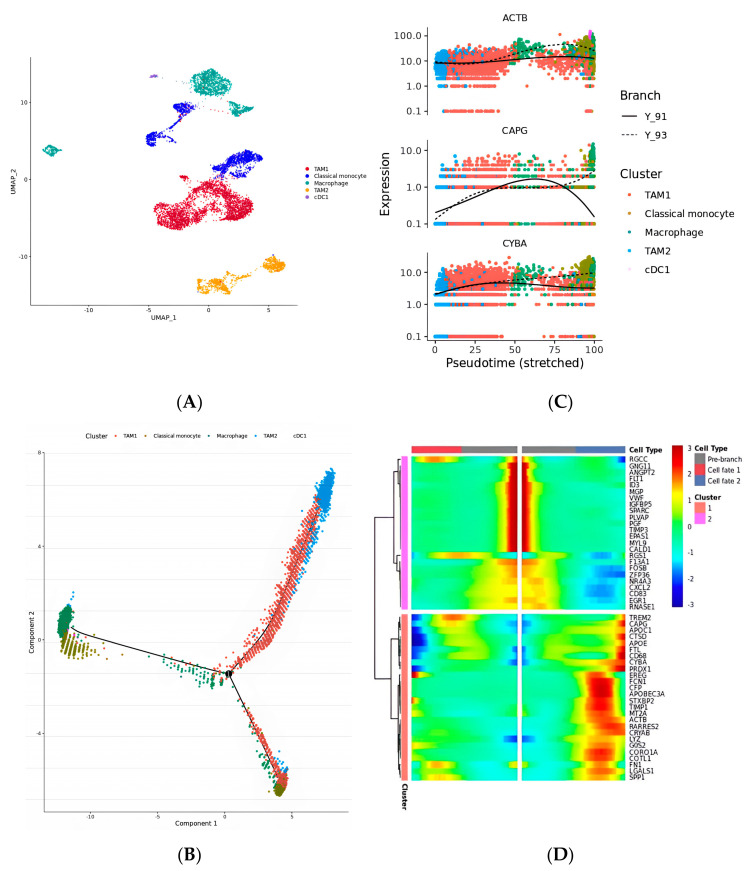
(**A**) UMAP of subtypes of macrophage cells. (**B**) Evolutionary trajectory of sub-clusters reconstructed the cell differentiation order. (**C**) Variate expression levels of hub genes in the development of the pseudotime trajectory. (**D**) Heatmap of top 50 genes which varied as a function of pseudotime.

**Figure 4 ijms-25-04134-f004:**
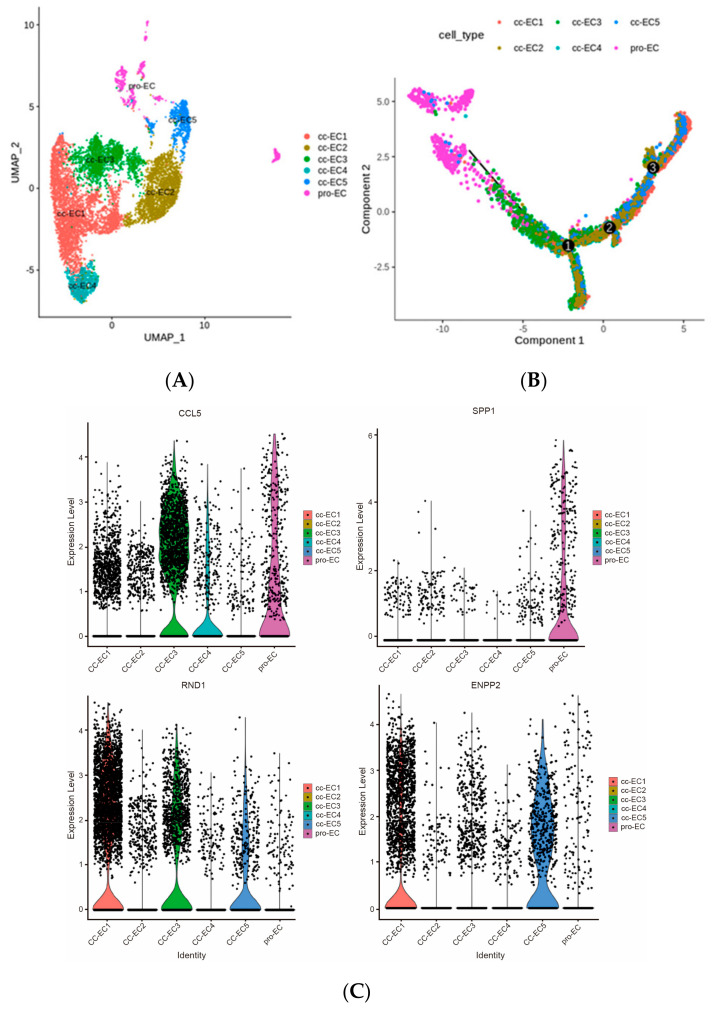
(**A**) UMAP clustering results of the subclusters of endothelial cells. (**B**) Evolutionary trajectory of the subclusters, reconstructing the cell differentiation order; numbers 1 to 3 denote the turning points of trajectory development. (**C**) The expression distribution of significantly expressed genes in different EC subtypes.

**Figure 5 ijms-25-04134-f005:**
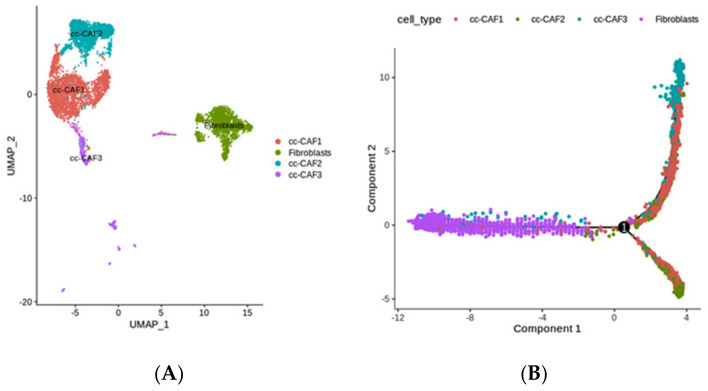
(**A**) UMAP of the subclusters of fibroblasts. (**B**) Evolutionary trajectory of subclusters according to their reconstructed cell differentiation order. (**C**) GO functional analyses of the subtypes CAF−2 and CAF−3, respectively.

**Figure 6 ijms-25-04134-f006:**
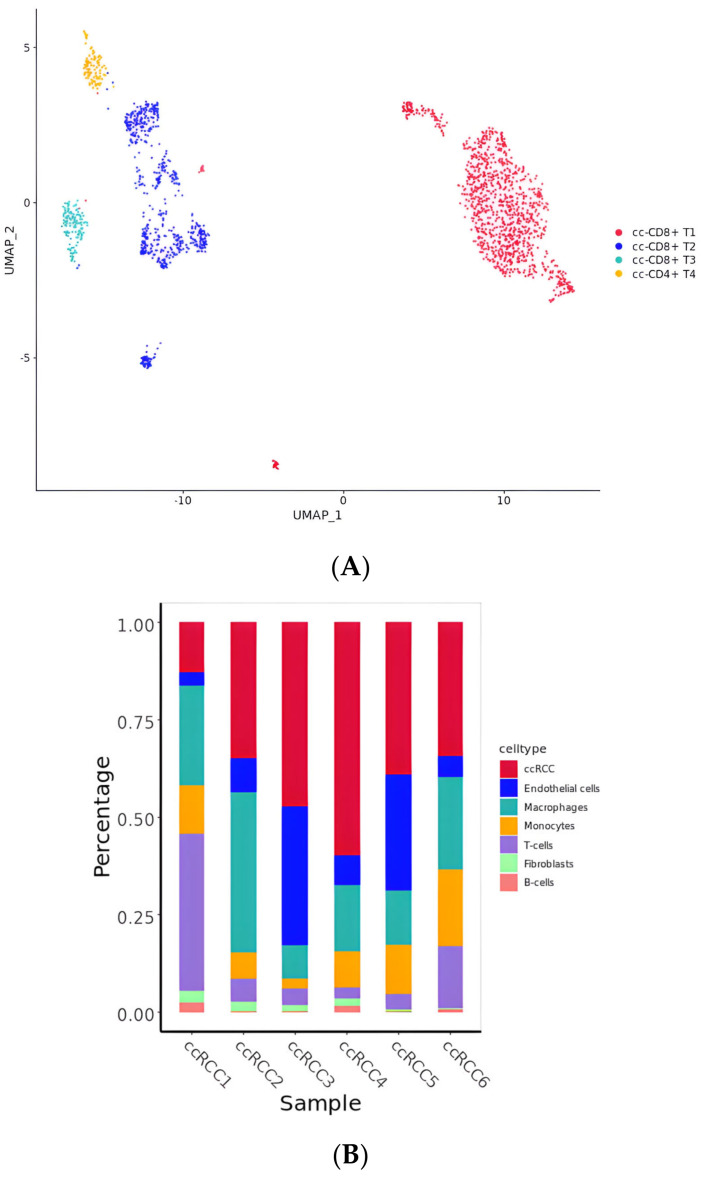
(**A**) UMAP of the subclusters in T cells. (**B**) The proportion of T cells in six ccRCC samples. (**C**) Survival analysis of patients stratified according to their expression of the marker genes of T cells in the TCGA dataset.

**Figure 7 ijms-25-04134-f007:**
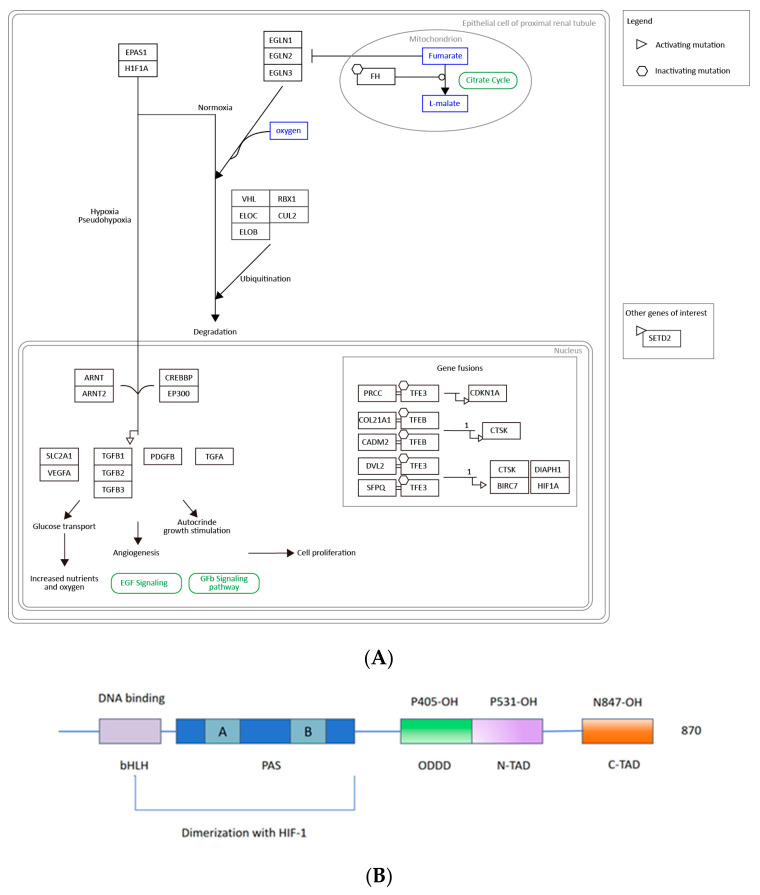
(**A**) The pathway that EPAS1 takes part in. (**B**) Structural domain of the protein EPAS1.

**Figure 8 ijms-25-04134-f008:**
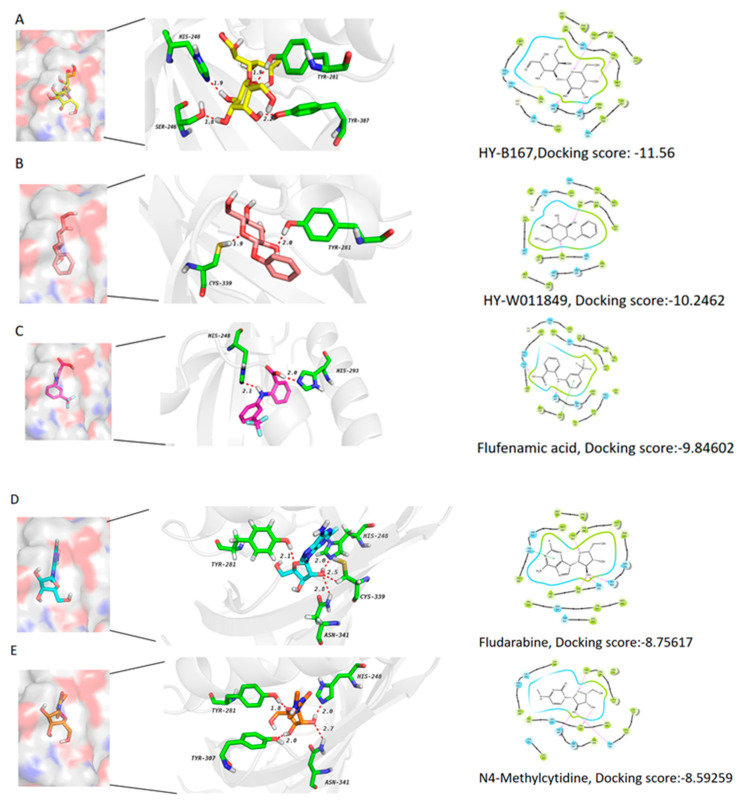
(**A**–**E**): docking models and chemical structures of the five screened compounds.

**Figure 9 ijms-25-04134-f009:**
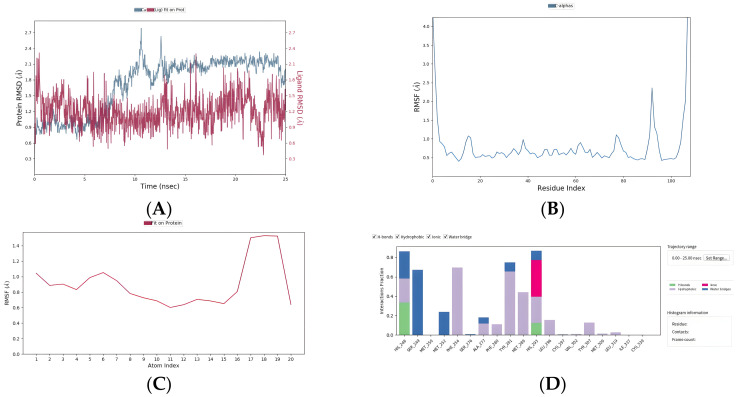
Molecular dynamics simulation results of FDA drugs flufenamic acid and fludarabine. (**A**–**D**) Flufenamic acid: root mean square deviation curve (RSMD), root mean square wave curve (RMSF), small molecule root mean square wave curve (RMSF), and statistics of interaction proportion of different residues. (**E**–**H**): The same metrics as above, but for fludarabine.

## Data Availability

All datasets used in this study were downloaded from the GEO (https://www.ncbi.nlm.nih.gov/geo, accessed on 20 January 2024) and MCE (https://www.medchemexpress.cn, accessed on 20 January 2024) websites.

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
