# Peer review of "Boosting Clear Cell Renal Carcinoma-Specific Drug Discovery Using a Deep Learning Algorithm and Single-Cell Analysis"

_ijms, 2024, doi:10.3390/ijms25074134_

Round 1

Reviewer 1 Report

Comments and Suggestions for Authors

First of all, I would like to thank you for inviting me to review the manuscript entitled: 'Deep learning algorithm and single cell analysis boosting the ccRCC specific drug discovery’. The general conclusion demonstrated that based on the differential typing of these key tumor microenvironment (TME) characteristics, and the computation of an integrated deep learning model, the pivotal transcription factor (TF) protein EPAS1/HIF 2α was identified, which regulates the expression of the genes involved in adjusting the mechanisms related to hypoxia, such as angiogenesis or apoptosis and tumor growth and invasion. Thus, I recommend publication after some major issues have been addressed:

Major:
1. Please avoid abbreviation in the title of the publication.
2. Please provide the general conclusion in the abstract.
3. The discussion should be much longer, especially, since the work contains a large number of results (one reference is insufficient). Please add a paragraph related to clinical and practical aspects of the study. Please provide explanation, how we can applicate your results into practice?, why your work is valuable in the field?

Minor:
1. The quality of the figures especially 3, 4, 5A, 5C, 7A should be improved.
2. Please provide limitations of the study.

Reviewer 2 Report

Comments and Suggestions for Authors

This study aimed to evaluate transcriptomic patterns associated with ccRCC tumor microenvironment (TME) components, based on available scRNA-seq data, to identify transcription factors related to ccRCC development, and eventually to select potential anti-ccRCC compounds from bioactive compound libraries, using deep graph neural network and machine learning algorithm. In general, the study is well designed, the methods are appropriate for this type of a study, and some new and significant results were obtained. There are several concerns that need to be clarified.

Firstly, it needs to be clear from the very beginning, i.e. from the abstract, that available scRNA-seq data from patients were used. The sentence “we used single cell transcriptome sequencing (scRNA seq) to describe the tumor microenvironment (TME)” may be misleading that you performed single cell transcriptome study.

Fonts in almost all figures need to be larger, some labels are not visible at all.

All authors of this study are from the School of Mathematics and Statistics, and there are no authors in the field of biomedical studies, which I consider very important when analyzing the activities of transcription factors etc.

ccRCC should be defined in the Title.

Line 21: FDA-approved drugs (better than compounds)

In Introduction, please specify that you refer to renal cell carcinoma (and not renal cancer in general) and check the epidemiological data (does RCC represent around 3% or 5% of all cancers?).

Line 71: Did you analyze protein structure of all 150 TFs? It should be clear in the text.

Line 75: Materials and Methods (instead of Methods and Materials)

Line 79: References for all studies referring to these single cell mRNA profiles of patients should be included, not only the database.

Lines 93-94: docking of ligands to the receptors (only ligands are being docked to the protein structure)

There are no line numbers after Figure 2.

Title of the part 3.8 should be more concise. Please, modify it.

DNMT inhibitor identified in your study should be specified in the text. It is not zebularine as it might be understood. Please, modify that part to be completely clear.

There is only one reference in Discussion. Please refer to the similar approach in the literature (maybe for some other diseases).

Conclusion: Please clarify how your approach may contribute to avoiding the drug resistance as stated.

Comments on the Quality of English Language

No major issues were detected.

Reviewer 3 Report

Comments and Suggestions for Authors

Dear authors,

Please address the following comments when you will revise the manuscript to amend its structure and quality. 

1- Please add all DOI identifiers to all cited literature. 

2- Please appropriately cite all protocols used in the M&M section. 

3- Please summarize the M&M section in a flowchart. 

4- I highly recommend the respected authors supplement all necessary data used in your analyses. 

5- Figures require further management. Please rearrange your figures and decrease the number of figures used within the paper. 

6- Figures' captions require clear explanations. Please improve your figures' captions details. 

7- Please supplement all R codes used in this study. I will check all applied codes separately to ensure that all steps were conducted correctly. 

8- What is the novelty of this study? and how your outcomes can help further studies in this area. Please add a brief explanation of the novelty of this study in the introduction section. 

9- Grammatical and typographical errors require further amendments. 

10- Figure 2 details have low resolution. Please improve the resolution of all figures used in this paper. 

11- Figure 3 caption is confusing. Please amend your explanations and add more details to these sections for all figures discussed within the text. 

12- Some of the represented figures can be supplemented. Please carefully redraft your paper. 

13- In the M&M section, please explain how the respected authors normalize the obtained data for their analyses. If specific procedures were used for this purpose, please clarify about them.

14- In Figure 8, right panel. The authors used the latest version of Schrodinger suite software. Since this software is a commercial suite the authors should clarify they have a valid license for using this software. If the cracked version was used to draw those figures represented in Figure 8, please remove all figures and try to use open-source tools such as PoseView or  PLIP (https://plip-tool.biotec.tu-dresden.de/plip-web/plip/index) or PlexView (https://playmolecule.com/PlexView/) to show your 2D plots. 

15- Discussion section: Please use updated papers to discuss and compare your results with the available literature. Please use active language to discuss the obtained results in this paper. 

16- Lines 81-86: Please supplement the SDF libraries of the mentioned datasets in this section. More importantly, based on which criteria do the respected authors filter the prepared datasets? I could not find any explanation of how the respected authors prepared their ligands for docking analyses. It was not mentioned how 3D geometries of applied ligands were energy minimized or optimized. Please add more details to this section. Anyway, as you know better, all docking results should be undergone molecular dynamics simulations. Please conduct at least 25 ns MD simulations for top hits discussed in the docking section of this paper and then discuss the results and the stability of the top best-docked poses within the cavity of target active sites. 

17- Please supplement all details related to Figure 7 either within the paper or the whole figure. 

18- Figure 5 section A: the authors should add clear explanations to this section. 

Generally, the manuscript has a weak draft and the authors must carefully revise the paper, amend grammatical errors, carefully manage the discussed figures, and completely discuss the methodology used for analyses of their data. The M&M section of this paper can be improved by adding more details to this section, supplementing necessary codes and raw data for further analyses, and increasing the reproducibility of the discussed data reported there. My recommendation for this paper is "Major revision" and I have no further comments at this step. 

Round 2

Reviewer 1 Report

Comments and Suggestions for Authors

I suggest to accept the manuscript in its current form.

Author Response

Thanks for your review.

Reviewer 2 Report

Comments and Suggestions for Authors

The authors improved their initial version of the manuscript and I support it to be published in the current form.

Comments on the Quality of English Language

No major issues were detected.

Author Response

Thanks for your review.

Reviewer 3 Report

Comments and Suggestions for Authors

Accept

Author Response

Thanks for your review.